# HIV and Substance Use in Latin America: A Scoping Review

**DOI:** 10.3390/ijerph19127198

**Published:** 2022-06-12

**Authors:** Hanalise V. Huff, Paloma M. Carcamo, Monica M. Diaz, Jamie L. Conklin, Justina Salvatierra, Rocio Aponte, Patricia J. Garcia

**Affiliations:** 1Department of Global Health and Population, Harvard T. H. Chan School of Public Health, Boston, MA 02115, USA; 2School of Public Health, Universidad Peruana Cayetano Heredia, Av. Honorio Delgado 430, San Martin de Porres, Lima 15102, Peru; paloma.carcamo.g@upch.pe (P.M.C.); patricia.garcia@upch.pe (P.J.G.); 3Department of Neurology, University of North Carolina at Chapel Hill, 170 Manning Drive, Campus Box 7025, Chapel Hill, NC 27599, USA; monica.diaz@neurology.unc.edu; 4Health Sciences Library, University of North Carolina at Chapel Hill, 335 South Columbia Street, Campus Box 7585, Chapel Hill, NC 27599, USA; jconklin@unc.edu; 5University Library Office, Universidad Peruana Cayetano Heredia, Av. Honorio Delgado 430, San Martin de Porres, Lima 15102, Peru; justina.salvatierra@upch.pe (J.S.); rocio.aponte.c@upch.pe (R.A.)

**Keywords:** Latin America, HIV/AIDS, substance use

## Abstract

This scoping review aims to explore the interplay between substance use (SU) and HIV in Latin America (LA). Database searches yielded 3481 references; 196 were included. HIV prevalence among people who used substances (PWUS) ranged from 2.8–15.2%. SU definitions were variable throughout studies, and thus data were not easily comparable. In 2019, only 2% of new HIV infections were attributed to injection drug use (IDU) in LA. Factors associated with HIV among PWUS included being female, IDU and homelessness, and PWUS were likely to engage in risky sexual behaviors, start antiretroviral treatment late, have poor adherence, have treatment failure, be lost to follow-up, have comorbidities, and experience higher mortality rates and lower quality of life, as has been reported in PLWH with SU in other regions. Five intervention studies were identified, and only one was effective at reducing HIV incidence in PWUS. Interventions in other regions have varying success depending on context-specific characteristics, highlighting the need to conduct more research in the LA region. Though progress has been made in establishing SU as a major concern in people living with HIV (PLWH), much more is yet to be done to reduce the burden of HIV and SU in LA.

## 1. Introduction

Substance use disorders (SUD) are maladaptive patterns of use of alcohol; caffeine; cannabis; hallucinogens; inhalants; opioids; sedatives, hypnotics and anxiolytics; stimulants; tobacco; and other substances leading to social, legal and occupational consequences [1]. In 2019, approximately 17 million people had SUD in Latin America (LA) and the Caribbean, leading to almost 24,000 deaths and more than 3 million disability-adjusted life-years [2]. Though illegal drugs are used more frequently in high-income countries, the disease, disability, and death experienced in low- and middle-income countries (LMIC) is disproportionate. Substance use (SU) is an immense concern for most countries within LA and deserves significant public health focus [3].

It is estimated that one in eight injection drug users (IDU) lives with HIV, amounting to 1.4 million people in the world. IDU are 22 times more likely than the general population to have HIV [4]. Beyond IDU, SU in general increases the risk of HIV infection. As of 2020, approximately 2.1 million people live with HIV/AIDS (PLWH) in LA [5]. Throughout the years, significant progress has been made toward the UNAIDS 90-90-90 goals: 80% of PLWH in the region know their serostatus, 81% of these are on antiretroviral therapy (ART), and 92% of these have achieved viral suppression, which has led to a 21% decline in AIDS-related deaths in LA between 2010 and 2020. However, the number of new HIV infections per year in the last 10 years has not changed [5]. In LA, the epidemic mostly affects men who have sex with men (MSM) and transgender women (TGW), but people who use substances (PWUS) remain a crucial vulnerable group and an understudied one [6].

The interplay between HIV and SU is complex and bidirectional: PLWH may be more likely than HIV-negative people to suffer from SU [7], and SU places individuals at higher risk of bloodborne transmission due to sharing of injection equipment, and of sexual transmission due to more engagement in high-risk behaviors [8]. Additionally, PLWH with SU may be at higher risk for adverse health outcomes [9,10].

Although SU clearly contributes to the burden of HIV worldwide, it is not easy to elucidate how this relationship plays out in the context of LA, which is diverse both culturally and geographically. In this scoping review, we aim to explore the published literature regarding the interplay between SU and HIV in LA.

## 2. Methods

The Preferred Reporting Items for Systematic Reviews and Meta-Analyses extension for Scoping Reviews (PRISMA-ScR) 2020 statement [11] was used to guide this review.

### 2.1. Eligibility Criteria

Studies were eligible for inclusion if they met the following criteria: (1) study in LA; (2) described the association between HIV and SU; (3) published as a full-text, peer-reviewed paper in English, Portuguese, or Spanish; and (4) published since 2012. This timeframe was selected to provide an accurate image of the state of current research in the region. Studies were ineligible if they did not include either PLWH or PWUS, or were published as case reports, editorials, commentaries, conference briefs, or review articles.

Eligible LA countries were Argentina, Belize, Bolivia, Brazil, Chile, Colombia, Costa Rica, Ecuador, El Salvador, Guatemala, Honduras, Mexico, Nicaragua, Panama, Paraguay, Peru, Uruguay, and Venezuela. We excluded studies from the Caribbean, Guyana, French Guiana, and Suriname.

We chose not to exclude studies based on their definition of SU. We use the term “substance” to refer to alcohol, tobacco, or recreational drugs, and “PWUS” to refer to those who use substances.

### 2.2. Information Sources and Search Strategy

The search strategy was developed in collaboration with librarians in the United States and Peru, including keywords and subject headings for three concepts: substance-related disorders, HIV, and LA. The complete, reproducible search strategy is available in Appendix A.

### 2.3. Study Selection

Search results were exported to Endnote X8 (Philadelphia, PA, USA) and duplicates were removed. Remaining studies were placed into Covidence (Veritas Health Innovation, Melbourne, Australia, available online www.covidence.org, accessed on 4 February 2022). In the initial title and abstract screening stage, two researchers (HVH, PMC) independently screened each reference for eligibility criteria. Conflicts were resolved by discussion between reviewers. In the full text review stage, one reviewer screened each article for eligibility. When a reviewer deemed an article ineligible, the second reviewer screened it, and a final decision was made through discussion. 

### 2.4. Data Extraction and Synthesis

For each study, one researcher independently extracted data. Unclear or missing information was discussed between two reviewers before making a final decision. The researchers synthesized data around themes identified during the extraction stage: prevalence of SU among PLWH, prevalence of HIV among PWUS, factors associated with HIV positivity among PWUS, risky sexual behaviors among PLWH and PWUS, health outcomes of HIV and SU, and risk reduction strategies for PLWH and PWUS.

## 3. Results

The PRISMA diagram is shown in Figure 1. Database searches yielded 3481 references, and 196 references were included. Table 1 shows the breakdown of studies by country.

### 3.1. Prevalence of SU in PLWH

We identified 83 studies that reported estimates of the prevalence of SU in PLWH in Brazil (59), Argentina (5), Peru (5), Mexico (5), Colombia (2), Ecuador (1), Venezuela (1), Chile (1), Guatemala (1), and Uruguay (1), plus two multinational studies. Study populations and definitions and measurements of SU differed significantly between studies. Papers estimated SU based on lifetime use, current use or using special screening tools (Table 2).

Estimates of alcohol use (AU) prevalence varied within Brazil from 7.8 to 80.2%, with the majority between 30–50% [12,13]. In Peru, AU prevalence was 30.7–54% [14,15], and in Guatemala 9.8% [16]. In Mexico, it ranged from 23% in pregnant PLWH in Mexico City to 73% in Jalisco [17,18]; and in Argentina, from 52.5% in TGW to 23% in prisoners [19,20].

The prevalence of illicit drug use (DU) in PLWH also varied significantly between and within countries with ranges of 4.5–76.5% in Brazil [21,22], 38% in Ecuador [23], 6–38% in Peru [15,24], 20.7–59% in Mexico [18,25], 8.4% in Venezuela [26], 10.5% in Guatemala [16], 15.9% in Chile [27], 7–62% in Colombia [28,29], and 46.4% in Uruguay [30]. Engagement with HIV care affects DU in Argentina, where 19% of PLWH reengaging in care and 76.7% of PLWH disengaged from care reported DU [31,32]. For example, following HIV diagnosis, less individuals reported DU or AU prior to or during sex in Peru and Colombia [33,34]. Similarly, testing positive for HIV at baseline was associated with IDU cessation in Mexico [35], less patients consumed alcohol after HIV diagnosis in Ecuador [36], and drug and tobacco use decreased after initiating ART in Brazil [37]. The incidence of IDU among PLWH in LA has declined over the decades following the start of the HIV pandemic [38,39]. 

The prevalence of smoking tobacco was between 12–80.8% in Brazil 119,100], 16.6% in Peru [14], between 20–60% in Mexico [16,17], 23.3% in Colombia [28], 50% in Venezuela [26], and 11.65% in Guatemala [20]. 

### 3.2. Prevalence of HIV in PWUS

Twenty-six studies estimated the prevalence of HIV among PWUS, with most studies from Brazil (10). HIV rates varied widely depending on the population characteristics, methodology, and geographic location of the study (Table 3)

HIV prevalence among PWUS ranged from 2.8 to 15.2% in Brazil [40,41], 3–4.8% in Tijuana, Mexico, where female sex workers (FSW), IDUs and MSM are concentrated [40,41,42,43,44,45], 7.7% in Ciudad Juarez, where syringe sharing is common among IDU [44], 2.0% among heroin users and 6.5% among syringe sharers in Colombia [46,47,48], and 13.7–34% in Argentina [49,50].

### 3.3. Factors Associated with HIV Positivity among PWUS

Being female has been associated with higher risk of HIV among PWUS in Brazil, Mexico, and Colombia [45,46,47,51,52]. A study in Tijuana found that sex work mediated 84.3% of the association between female gender and HIV incidence [48]. In Brazil, HIV risk in females was mediated by high levels of unprotected sex, gender-based violence, and frequent unsafe injection habits [49]. However, several Colombian studies and one from Brazil failed to show significant differences in HIV rates between male and female IDU [50,53,54,55].

SU has also been associated with vertical transmission of HIV. A Brazilian study of pregnant PLWH showed a significant association between IDU and vertical transmission, increasing the odds of transmission by 11% [56]. Those who use substances were less likely to attend prenatal visits, use ART, and have an undetectable viral load (VL) close to birth and their HIV-exposed children were more likely to be lost to follow-up [57,58,59].

IDU seems to confer a higher HIV risk than non-IDU in Brazil [49,60]—for instance, a study conducted among DU in northern Brazil found an HIV prevalence of 27.8% among IDU and 13.1% among non-IDU [50]. This may be mediated by syringe sharing, an HIV risk factor reported in Ecuador, Colombia, and Mexico [43,55,61]. Among Colombian IDU, sharing needles and/or syringes led to an infection risk 5.07 times higher than for those who did not share [55]. Additionally, syringe confiscation by police was positively associated with testing positive for HIV in Mexico and similarly, difficult access to sterile needles was associated with elevated injection risk [62,63].

Other deleterious practices include not disinfecting syringes, having multiple injection partners, sharing drug mixtures, and being injected by a person who charges to inject [45,61,64]. More frequent DU and longer history, as well as having severe problems with DU, may also increase the risk of HIV [36,50,65,66]. 

Homelessness among PWUS has also been associated with higher HIV rates in Brazil [43,67]. A Mexican study found that more hours on the street per day was spatially correlated with HIV infection in these populations [45].

### 3.4. Risky Sexual Behaviors among PLWH and PWUS

Sexual behaviors that lead to higher HIV risk, such as engaging in condomless sex or having multiple sexual partners, are more frequently described among PWUS than in the general LA population [40,43,47,50,60,68,69]. A Brazilian study showed that PWUS had a probability of engaging in high-risk sexual behaviors 3.64 times greater than non-substance-users [66,70]. Inconsistent condom use has been associated with DU in Brazil [47,66,68,71,72,73,74], Costa Rica [75], Colombia [53,54,65,76], Mexico [40,77], Chile [69], El Salvador [78,79], and Guatemala [80]. In regions with high rates of sex work, these associations might explain the link between female sex, HIV and DU.

Some drugs may be associated with more risky sexual behaviors than others. In Colombia, cocaine use was associated with sharing drug mixtures and less condom usage [64]. In Brazil, high-risk sexual behaviors were more likely in users that snorted cocaine versus those than smoked cocaine [66,81]. More risky sexual behaviors were found in heroin-users in Brazil [82], heroin plus methamphetamine users in Mexico [83], and polydrug or poly route SU in Mexico [84].

PWUS are likely to use substances right before sex, thereby increasing risk of HIV infection and transmission: 62.1% and 59.1% of Brazilian crack users reported AU or DU, respectively, before sex [68]. Other studies report similar findings [71,73]. Using substances before sex may lead to reduced condom use by almost 68% [73], increased sexual promiscuity [85], and less likelihood of serostatus disclosure [86]. The correlation between SU before sex and inconsistent condom use was also described in serodiscordant couples [86], IDUs who share needles [54], and PLWH who use cocaine [64].

SU has also been associated with transactional sex in Chile [69], Mexico [41], Brazil [49,71,87], El Salvador [79] and Peru [88]. In a Brazilian study, 12.5% of crack users reported engaging in sex work, and 10% reported using sex to obtain drugs [87]. 

### 3.5. Health Outcomes of PLWH Who Use Substances


Late diagnosis and treatment initiation


A late HIV diagnosis can lead to worse response to treatment, higher mortality risk, and increased risk of onward transmission [89], and has been associated with DU in Peru, Uruguay and Argentina [30,90,91]. In Argentina, high levels of AU, but not DU, were significantly associated with delayed ART initiation [92]. Delays may be caused by providers’ belief that ART is contraindicated in DU, as was found in a qualitative study in El Salvador [79]. 


Treatment adherence (TA) and treatment failure (TF)


Several studies in LA have reported associations between SU and poor TA, and consequently TF [24,93,94,95,96,97,98,99,100,101,102,103,104,105,106,107,108,109,110,111,112,113,114,115,116,117,118]. In Brazil, 29.3% of PLWH reported not taking ART because of SU [94]. Both Brazilian PLWH and their providers shared concerns about the interactions between drugs, alcohol and ART with some providers recommending stopping ART during SU [79,97]. Furthermore, AU has been found to increase the risk of hepatotoxicity associated with ART [13,119]. Nonetheless, several studies have found no association between SU and poor TA [12,29,101,102,113,118,120,121,122,123,124].

SU has also been associated with TF as evidenced by virologic failure (VF) in Brazil [37,99,113,119,120,125], Guatemala [16], Peru [15,121], and a multinational study [122]. In Brazil, PLWH with a detectable VL had 1.76 times higher chance of AU [120], while DU increased risk of a detectable VL by 3 times [119]. AU by caregivers was associated with detectable VL among Brazilian children and adolescents with HIV [125]. However, a few studies did not find an association between SU and viremia [123,124]. Other studies in Brazil found that SU was correlated with lower CD4-cell counts, another measure of TF [123,126,127], and shorter time to progression to AIDS [128]. 


Loss to follow-up (LTFU)


PLWH who use substances are also more likely to be LTFU. Argentinian providers reported that IDU had high rates of LTFU, missed appointments, and treatment nonadherence [129]. Additionally, DU among Brazilian HIV-positive mothers was associated with LTFU for their at-risk children [57]. AU also correlated with LTFU, though other SU did not [28,130]. 


Comorbidities, quality of life and mortality in PLWH with SU


SU can be considered a comorbidity in PLWH. Among Brazilian PLWH, smoking tobacco was the most common comorbidity [131]. SU in PLWH is also associated with several communicable and non-communicable comorbidities, as listed in Table 4.

SU has also been linked to lower quality of life [21,132,133,134,135,136,137], poor sleep quality [25], discrimination [138] and gender identity stigma [139], although not with food insecurity [140].

Higher mortality rates among PLWH in Brazil [141,142,143,144,145,146,147] and Mexico [44] have been associated with SU. Brazilian PLWH using drugs had a relative mortality risk of 3.1 compared to non-SU [143]. However, history of DU in Uruguay was not associated with dying of AIDS [30]. Another study in Brazil found that although heavy AU and tobacco were significantly associated with death; light AU was protective [146].

### 3.6. Strategies for Reducing HIV-Risk in PWUS in LA


Educational interventions


We identified one randomized factorial trial conducted in Tijuana and Ciudad Juarez, Mexico, among FSW-IDUs, where participants underwent an interactive or didactic safe-sex educational intervention and an intervention to reduce needle sharing, after which they were interviewed and tested for HIV on a quarterly basis. One year following the intervention, the HIV/STI incidence decreased by more than 50% in groups that had received the interactive safe sex intervention in both cities. Additionally, when didactic injection risk interventions were coupled with expanded access to sterile injection equipment, there were significant reductions in needle sharing [148]. These findings underscore the importance of educational interventions that go beyond a standard lecture format, and the critical role of ensuring access to sterile syringes.


Peer-referral networks for HIV testing among PWUS


In El Salvador, an interrupted-time series study was conducted to assess the effect that peer-referral chains had on HIV testing among crack users. Participants compiled a list of crack users with HIV risk within their social network and were given coupons to refer them to HIV testing at a community site. This strategy was found to increase monthly numbers of HIV crack-using testers fourfold, but did not affect the HIV incidence [149], highlighting that although testing may be a critical component of HIV prevention, it is not sufficient. In the same study, a multi-level, community-based prevention intervention involving rapid HIV testing, social network referral systems, and peer small group educational sessions was carried out. Exposure to the interventions and increasing HIV testing led to reductions in condomless sex, but not in the full study sample, and changes in the HIV incidence were not mentioned [150].

### 3.7. Strategies for Reducing SU Risk in PLWH in LA


Medication-Assisted Therapy (MAT) for SU treatment


In Peru, a double-blind randomized controlled trial including 155 PLWH with alcohol use disorder (AUD) found that naltrexone, an effective aid in SUD treatment, was safe to use concurrently with ART, with no significant difference in adverse events in the treatment and placebo groups [151]. These findings are especially relevant in the context of LA, where, as previously described, some healthcare providers believe that ART is contraindicated in PWUS.


Trans-sensitive healthcare


A cohort study was conducted among TGW initiating ART in a trans-sensitive clinic in Argentina. The clinic had essential components including using patients’ preferred names and pronouns, having trans-competent trained providers, integrating multiple relevant services, adjusting to relevant social contexts, and including peer navigators that assist patients and link them with services. After 6 months of care, a significant reduction in SU was found [139]. Despite the lack of a control group in this study, these results highlight the potential that tailoring health services to at-risk populations may have on SU among PLWH.

## 4. Discussion

Through review of the included studies, several recurring themes became apparent regarding HIV and SU: unequal representation of countries in LA, heterogeneity in measurements of data, and a relative abundance of publications on prevalence, risk and outcomes as compared to interventions.

The prevalence of SU among PLWH worldwide is reported to be higher than in general populations. In this review, most papers describing this prevalence were from Brazil, with some papers from Peru, Mexico, Colombia, and Argentina, and many countries with one or zero papers. PLWH SU prevalence ranges were wide. The overall prevalence range for AU was 7.8–80.2%, for smoking 11.7–80.8%, and for DU 4.5–76.5%. In the United States, 27% of PLWH have AU, 33.6% smoke tobacco, and 36% report DU, with the most frequently used being marijuana, methamphetamines, and cocaine/crack [152,153,154]. The estimated average 1-year prevalence of AUD among PLWH in Africa was 20.03%, ranging from 16.6% in Uganda to 28.8% in South Africa [155]. Comparison of the ranges reported in our review with prevalence estimates from other regions is challenging, as the included studies lacked consistency in SU definitions. Furthermore, because not all studies identified the specific substance used, but rather categorized many as “illicit” or “recreational” drugs, establishing the most frequently used substances within LA is difficult. Additionally, the large variation in prevalences could also be attributed to the heterogeneity of cultures, geography, and economics of the various regions of LA. 

The Pan American Health Organization (PAHO) and the World Health Organization have published recommendations for improving comparability of SU research studies including using standardized drug categories, using three time-periods to measure prevalence of use, asking about age at first use, routes of administration and consequences of use, and disaggregating data by sex [3,156]. Most reviewed studies did not follow these recommendations, despite being published after the recommendations were released. For instance, some studies did not disaggregate data by sex, or by type of drug use. Furthermore, a review in 123 LMIC intending to assess the availability and quality of HIV prevalence data among at risk groups (MSM, IDU, FSW and TGW) found that only 14.6% of countries had prevalence data on all four risk groups, yet it is estimated that 62% of new HIV infections occur among these populations and their partners [157,158]. Without quality data, it becomes difficult to understand the burden of the problem, develop interventions and measure their success. 

Risk factors for HIV among PWUS in the LA literature included being female, IDU, unclean injection practices, and homelessness. Higher HIV risk among female IDU was explained by an overlap between DU and sex trade in one Mexican study [48] and partly by gender-based violence in a Brazilian study [49]. In contrast, stigma, physical and sexual violence, mental illness, social marginalization, and economic vulnerability have been identified in other regions as determinants of HIV infections among PWUS [159]. 

In 2019, globally, 10% of new adult HIV infections happened among IDU, though the percentage varied greatly by region. In eastern Europe and central Asia, 48% of new HIV infections were attributed to IDU as opposed to only 2% in LA [157]. In LA, the risk for new HIV infections is largely associated with risky sexual behaviors including condomless sex, having multiple partners, SU directly before sex and transactional sex. Vertical transmission of HIV has also been linked to maternal IDU, as previously reported [160].

Poor clinical outcomes are linked to SU among PLWH in LA, including late diagnosis and treatment initiation, poor TA, TF, LTFU, and numerous comorbidities. These findings mirror those from elsewhere in the world. In North America, PWUS are also less likely to be linked to HIV care, be retained, be adherent to ART, and achieve viral suppression [161,162,163,164]. Notably, we found that both PLWH and health providers in LA expressed concerns about interactions between ART and SU, and some providers even recommended stopping ART at times of SU. Despite a higher risk of toxicities and adverse events in PLWH with SU [165], ART should not be suspended even in the context of SU due to risks of viral rebound, immune decompensation or clinical progression [166]. Comorbidities among PLWH with SU in other regions have also been reported in the literature, including infectious [167,168,169] and noninfectious diseases [170,171,172].

In this review, only three papers described strategies to reduce HIV risk among PWUS in LA. One focused on educational interventions, the other two focused on increasing HIV testing through peer referral chains, but only the first was successful in reducing HIV incidence. The other two measured surrogate outcomes such as HIV testing and condomless sex, which did not translate into changes in HIV incidence. Studies in other regions have found Pre-Exposure Prophylaxis (PrEP) to significantly reduce HIV acquisition in IDU and to be well-accepted among PWUS [173,174]. Unfortunately, no studies were found that evaluate PrEP or Post-Exposure Prophylaxis (PEP) use in PWUS in LA. 

HIV prevention studies among PWUS outside of LA include behavioral psychosocial interventions to reduce sex and drug risks, social service interventions, opioid antagonist therapy, financial and scheduling incentives, and syringe service programs to improve access to sterile injection equipment [175]. These interventions have varying degrees of success in reducing risky behaviors and HIV incidence depending on the populations and specific methodology [175], underlining the importance of researching their effectiveness in the context where they are to be applied. 

Only two studies focused on strategies to reduce SU risk among PLWH in LA. One found trans-sensitive clinics successful, albeit using a cohort study design, and the other found concurrent ART-MAT use to be safe. Globally, interventions to reduce SU among PLWH include opioid antagonist therapy, behavioral or psychological interventions, and integrated HIV-SU services [176,177,178,179]. Once again, the effectiveness of these interventions is variable throughout studies, and generating strong evidence for the specific context of LA is fundamental.

Some themes, like pathophysiology or mechanisms of SUD in PLWH are well described in other regions, but are not present in the LA literature. For example, some US studies look at the long-term neurocognitive consequences of chronic cannabis use [180], while other studies even look into the beneficial properties of certain substance such as smoked medical cannabis for neuropathic pain [181]. The contexts as well as priorities for HIV research on SU therefore vary greatly depending on the region. Researchers in LA may face unique challenges such as population heterogeneity even within countries, competing public health priorities, and potential lack of funding and research infrastructure, which may explain the gaps in the literature. 

PAHO identified three factors that lie at the heart of the problem of SU within LA: “inequities in development, lack of access to health services and the exclusion of some segments of the population” [3]. Collaborative efforts to bridge the gaps in knowledge and outcomes will lead to generation of better, meaningful data which can lead to policy changes and better patient outcomes. Despite gaps in information on certain countries and the challenge in comparability of other data, it is clear that SU plays a role in the HIV/AIDS pandemic as it affects LA and much more research needs to be conducted to provide equitable solutions.

**Table 2 ijerph-19-07198-t002:** Prevalence of substance use (SU) among people living with HIV (PLWH) in Latin America.

Country	Population	Definition of SU	Prevalence of SU	Study
Brazil	PLWH (Bahia)	Alcohol use, current alcohol use	42.4%, 38.5%	[99,100]
		Illicit drug use	4.9–11.5%	
		Alcohol and non-injectable illicit drug use	11.8%	
		Smoking	80.8%	
	PLWH with TB (Ceará)	Smoking	33.5%	[182]
		History of alcoholism	42.0%	
		History of illicit drug use	26.5%	
	PLWH (Ceará)	Alcohol use disorder	40.0–49.5%	[140,183]
		History of alcohol consumption	44.8%	
		Alcohol use	20.5%	
		Alcohol dependence	19.0%	
		Risky alcohol consumption	10.5%	
		Illicit drug use	19.3%	
		Drug use	12.7%	
	PLWH with HBV (Ceará)	IDU	9.1%	[184]
		Inhaled cocaine use	12.0%	
		Alcohol use	36.0%	
	PLWH with HCV (Ceará)	IDU	22.2%	
		Inhaled cocaine use	16.7%	
		Alcohol use	44.4%	
	PLWH (Espírito Santo)	Smoking	22.6%	[21]
		Alcohol use	32.8%	
		Illicit drug use	4.5%	
	PLWH (Goiás)	Current, former smoking	24.1%, 23.8%	[185]
		Risky consumption of alcohol	71.4%	
	PLWH (Goiás)	Alcohol use	41.8%	[114,186]
		Illicit drug use	13.2–25.7%	
	PLWH (Minas Gerais)	Hazardous alcohol use (male, female)	8.6%, 16.0%	[103]
	PLWH (Minas Gerais, Rio de Janeiro)	Previous smoking	28.3%	[187]
		Current smoking	37.6%	
		Past alcohol use	35.0%	
		Current alcohol use	57.7%	
		History of illicit drug use	52.5%	
	Pregnant women with HIV with drug use history (Minas Gerais)	Drug use during pregnancy	7.6%	[58]
		Smoking during pregnancy	52.9%	
		Alcohol use during pregnancy	30.6%	
		Smoking and alcohol use during pregnancy	18.8%	
		Drug use prior to pregnancy		
		Cocaine	16.7%	
		Crack	43.4%	
		Cocaine and crack	8.3%	
		Marijuana	15%	
		Cocaine, crack and marijuana	8.3%	
		Other types of drugs	8.4%	
	PLWH (Minas Gerais)	Alcohol use	40.2–80.2%	[13,188,189]
		Smoking	22.8–26.1%	
		Illicit drug use	4.4–50.3%	
	PLWH with TB (Minas Gerais)	Alcoholism	30.7%	[190]
		Illicit drug use	23.5%	
		Smoking	26.8%	
	PLWH (Pernambuco)	Smoking	28.9–54.7%	[109,127,191]
		Alcohol use	35.6%	
		Illicit drug use Marijuana Cocaine Crack Glue	27.6% 26.6% 9.1–9.9% 6.5–6.7% 5.8%	
	PLWH with pulmonary TB (Pernambuco)	Illicit drug use	30.9%	[192]
	PLWH (Rio de Janeiro)	Smoking (lifetime, last 3 months)	23.9–55.3%, 21–20.9%	[135,193,194]
		Alcohol use (lifetime, last 3 months)	23.1%, 34.3%	
		Marijuana (lifetime)	23.1%	
		Cocaine (lifetime)	20.7%	
		Polysubstance (last 3 months)	2.4%	
	PLWH with MDR TB (Rio de Janeiro)	Illicit drug use	19.2%	[195]
	Women living with HIV (Rio de Janeiro)	Current or past smoking	42.4%	[196]
		Lifetime illicit drug use	16.6%	
	Men living with HIV (Rio de Janeiro)	Alcohol misuse	34%	[113]
		Non-IDU	76.7%	
	Pregnant women with HIV (Rio de Janeiro)	Illicit drug use (before, during pregnancy)	18%, 6%	[95]
		Alcohol use (before, during pregnancy)	51.3%, 14%	
		Smoking (before, during pregnancy)	33%, 15%	
	PLWH (Rio Grande do Norte)	Smoking	12%	[197]
		Alcohol use	29%	
		Illicit drug use	8%	
	Hospitalized PLWH (Rio Grande do Norte)	Smoking	41%	[98]
		Alcohol use	51%	
		Illicit drug use during week of admission	31%	
	PLWH (Rio Grande do Sul)	Propensity for alcoholism	37.5%	[12,94,123,198,199,200]
		Possible alcohol dependence	5%	
		Alcohol (weekly use, use, harmful use, abuse)	31.1%, 58.4%, 7.9%, 28.6%	
		Alcohol or drug use	7.8%	
		Inhaled drug use	33.1%	
		IDU	13.9%	
		Smoking	54.7%	
	People with AIDS (Rio Grande do Sul)	IDU	12.2%	[39]
	PLWH attending Special Assistance Services (Rio Grande do Sul)	Smoking (abuse or addiction)	45.6%	[134]
		Alcohol abuse or addiction	32.7%	
		Other substance abuse or addiction	15.7%	
	PLWH with TB (Rio Grande do Sul)	Alcoholism	25.7–44.0%	[201,202]
		Smoking	40%	
		Illicit drug use	37.5%	
	Women living with HIV (Rio Grande do Sul)	History of drug use	29.8%	[203]
	Mothers living with HIV (Rio Grande do Sul)	Drug use	28.9%	[59]
	PLWH with oral lesions (Rio Grande do Sul)	Smoking	30.7%	[204]
		Illicit drug use	17.2%	
		Alcoholism	14.4%	
	PLWH (Santa Catarina)	Smoking, current smoking	45.9%, 32.1%	[131,205]
		Alcoholism, alcohol use	13.3%, 31.1%	
		Illicit drug use	10%	
	PLWH (São Paulo)	Alcohol use in last month (any amount, >1 time/week)	50%, 16.9%	[73,120,126,206,207]
		Alcohol use, risky use, harmful use, abuse, dependence	40.6%, 14%, 12.6%, 18.3%, 5.5%	
		Illicit drug use (in the last month)	9.3–10%	
		SU (last month, last year)	38%, 62%	
		Alcohol use during sex	42.6%	
		Drug use during sex	19.6%	
	PLWH without AIDS (São Paulo)	IDU (Pre-ART period, post-ART period)	21%, 8%	[128]
	LGBT PLWH (São Paulo)	Drug use	76.5%	[208]
	Women living with HIV (São Paulo)	Crack use	3%	[22]
		Other drug use	10%	
	TGW living with HIV (São Paulo)	Illicit or recreational drug use	46%	[132]
	PLWH (multicentric)	Abusive use of alcohol (Recife, Goiania, Porto Alegre)	22.8%, 26.3%, 5.6%	[115]
		Current smoking (Recife, Goiania, Porto Alegre)	24.4%, 23.0%, 42.3%	
		Lifetime use of crack (Recife, Porto Alegre)	6.0%, 8.9%	
		Lifetime use of cocaine (Recife, Goiania, Porto Alegre)	9.0%, 10.7%, 29.6%	
	PLWH on ART (multicentric)	IDU (male, female, starting ART in 2006, starting ART in 2015)	3%, 0.6%, 6.8%, 1,4%	[142]
		Non-IDU (female)	23%	
	Women living with HIV (multicentric)	Smoking	19.6%	[209]
		Illicit drug use	18.7%	
	PLWH (nationwide)	Alcohol use	49.5%	[210]
		Smoking	45.3%	
		Amphetamines	1.7%	
		Marijuana	10.5%	
		Powder cocaine	3.6%	
		Crack cocaine	5.3%	
		Inhalants	3.6%	
		Ketamine	1.7%	
		Opioids	1.7%	
Ecuador	Newborns with HIV (Babahoyo)	Mothers using parenteral drugs	38%	[23]
Peru	PLWH (Lima)	Current smoking	16.6%	[14]
		Current alcohol use, pathological use	30.7%, 3.4%	
		Marijuana use (past, current)	11.7%, 1%	
		Past cocaine paste use	10.2%	
		Past cocaine use	7.8%	
		Other past drug use (crack, poppers, terokal)	1.4%	
	TGW living with HIV (Lima)	Hazardous alcohol use	40%	[15]
		Harmful alcohol use	2%	
		Alcohol dependency	12%	
		Low drug use severity	28%	
		Moderate drug use severity	4%	
		Substantial or severe drug use severity	6%	
	PLWH from tertiary care center (Lima)	Illicit drug use	6.9%	[90]
	MSM and TGW living with HIV (Lima)	Alcohol use disorder	28.0–43.2%	[24,211]
		Alcohol dependence	3.9%–5.3%	
		Recent drug use	6.0%	
		Low drug use severity	20.2%	
		Moderate drug use severity	5.3%	
		Substantial drug use severity	1.7%	
Mexico	PLWH (Jalisco)	Smoking	60%	[18]
		Alcohol use	73%	
		Drug abuse	59%	
	PLWH (San Luis Potosi)	Alcohol use (Male, Female)	45%, 20%	[106]
		Drug use (Male, Female)	21%, 3%	
	PLWH (Mexico City)	Illicit drug use	20.7%	[25]
		Nicotine use	24.3%	
	Recently pregnant women with HIV (Mexico City)	IDU, cocaine or heroin use	0%	[17]
		Marijuana use	2%	
		Inhaled solvent use	26%	
		Current smoking	20%	
		Alcohol use in last 6 months	23%	
	PLWH (Puebla)	Drug or alcohol addiction	30.6%	[112]
Venezuela	PLWH (Valencia)	Inhaled drug use or IDU	8.4%	[26]
		Smoking 0.5 pack/year	50%	
		Smoking 0.15 pack/year	16%	
Chile	PLWH (Araucanía, Metropolitan)	Drug use (Mapuche, other ethnicity)	8.6%, 17.2%	[27]
Colombia	PLWH (Bogota)	Alcohol use		[28]
		≤1 time/month	55.8%	
		2–4 times/month	14.0%	
		2–3 times/week	2.3%	
		Smoking	23.3%	
		Drug use	7.0%	
	PLWH from a tertiary care center	Psychoactive substance use	62%	[29]
Argentina	Prisoners living with HIV (Buenos Aires)	Hazardous alcohol use	23%	[20]
		Alcohol dependence	39%	
		SU in 30 days prior to incarceration		
		Cocaine	46%	
		Crack	46%	
		Opioids	2%	
		Benzodiazepines	19%	
	TGW with HIV initiating ART (Buenos Aires)	Hazardous alcohol use	52.5%	[19,139]
		Drug abuse	13.1%	
		Cocaine use in past year	52.5%	
		Marijuana use in past year	54.1%	
	PLWH disengaged from HIV care (multicentric)	SU in last 6 months (TGW, cisgender)	73.2%, 24.2%	[32]
		Cocaine in last 6 months (TGW, cisgender)	76.7%, 24.2%	
		Substance-related problems (TGW, cisgender)	39.0%, 10.3%	
		Hazardous alcohol use (TGW, cisgender)	46.3%, 23.1%	
	PLWH reengaging in HIV care (multicentric)	Substance abuse	19%	[31]
Guatemala	PLWH attending an Infectious Diseases clinic (Guatemala City)	Smoking	11.65	[16]
		Excessive alcohol consumption	9.8%	
		Prior illicit drug use	10.5%	
Uruguay	PLWH who died from AIDS (nationwide)	History of drug use	46.4%	[30]
		History of IDU	11.4%	
Multi-country	PLWH (Buenos Aires, Argentina; Rio de Janeiro, Brazil; Santiago, Chile; Tegucigalpa, Honduras; Mexico City, Mexico; Lima, Peru)	>3 drinks of alcohol in last 7 days	13.7% (4.9–23.9%)	[118]
		Marijuana use in last 7 days	4.1% (1.9–11.1%)	
		Cocaine use in last 7 days	1.4% (0.7–3.5%)	
		Crack use in last 7 days	0.3% (0.0–3.3%)	
		Heroin use in last 7 days	0.1% (0.0–0.6%)	
		Any illicit drug use in last 7 days	5.0% (2.8–14.0%)	
	PLWH (Buenos Aires, Argentina; Rio de Janeiro, Brazil; Santiago, Chile; Tegucigalpa, Honduras; Mexico City, Mexico; Lima, Peru)	Alcohol use only	26.2%	[130]
		Non-IDU only	2.1%	
		Alcohol + non-IDU	3.5%	

**Table 3 ijerph-19-07198-t003:** Prevalence of HIV among people who use substances in Latin America.

Country	Population	Prevalence of HIV	Study
Brazil	Crack cocaine users in a referral hospital (Goiás)	2.8%	[51]
	Non-IDU (Goiás)	3.2%	[81]
	Crack users (Rio/Salvador)	3.7%/11.2%	[87]
	Illicit drug users (Northern Brazil)	15.2%	[50]
	Drug users (Recife)	5.3%	[47]
	Drug users (São Paulo)	5.8%	[60]
	Migrant heroin users receiving treatment (São Paulo)	12%	[82]
	Polydrug users (multicenter)	5.8%	[49]
	Drug users (multicenter)	6%	[212]
	Crack users (multicenter)	4.3%	[67]
Mexico	Drug treatment center clients (West Central Mexico)		[213]
	Community	1.6%	
	In-prison	6.7%	
	IDU (Tijuana)	3–4.4%	[41,42,43,44]
	Low-risk non-IDU (Tijuana)	3.7%	[40]
	IDU		[42]
	Hermosillo	5.2%	
	Ciudad Juárez	7.7%	
Colombia	IDU		[76]
	Pereira	1.9%	
	Cali	2.2%	
	Bogotá	3.0%	
	Cúcuta	6.7%	
	Heroin users (Pereira, Medellin)	2.0%	[53]
	IDU (Medellin)	3.6–3.8%	[55,76]
	IDU (Armenia)	2.6–2.7%	[65,76]
	IDU (multicenter)	4.8–6.5%	[52,214,215]
Argentina	Current or former drug users from rehabilitation center and HIV clinic (Buenos Aires)	34%	[216]
	People who have sex under the influence of substances (multicenter) Chemsex users (multicenter)	13.7% 23.3%	[217]

**Table 4 ijerph-19-07198-t004:** Comorbidities associated with substance use among people living with HIV.

Non-Infectious	Infectious
Cardiovascular	Bacterial
High blood pressure [194,218]	Pneumonia [197,219]
Strokes, cardiovascular accidents [194,197]	Tuberculosis [190,195,220]
Endocrine	Treatment default [190,221]
Diabetes mellitus [194]	Treatment delay [192]
Early natural menopause [222]	Treatment failure [195]
Nutritional	Loss to follow-up [195]
Malnutrition [200]	Mortality [221]
Low body mass index [200]	Multi drug resistance [221]
Low skeletal muscle mass index [200]	*Chlamydia trachomatis* urethritis [223]
Osteopenia and osteoporosis [188]	Syphilis [22]
Gastrointestinal symptoms [185]	Viral
Malignancies	HTLV-1/2 [224,225]
Lung cancer [197]	Hepatitis B [186,226]
Kaposi’s sarcoma [227]	Hepatitis C [215,216,228,229,230,231,232]
Anal intraepithelial neoplasia [187]	Hepatitis E [198]
Cervical cancer and HPV [209,233,234]	Epstein-Barr virus [39,235]
Psychiatric	Fungal
Depression [14,19,24,112,126,194,197]	Oral candidiasis [235,236]
Suicidal ideation [31,60]	
Anxiety [189]	
Bipolar disease [237]	
Manic symptoms [183]	

## 5. Conclusions

The interplay between SU and HIV in LA is complex and bidirectional. Numerous studies present prevalence of SU within PLWH though with inconsistent definitions rendering the data difficult to analyze and compare with that of other regions. The data do, however, indicate that SU is a persistent problem of particular concern within PLWH in LA. The issue of SU within PLWH carries grave importance given numerous associated communicable and non-communicable comorbidities, as well as lower quality of life and higher mortality risk within this population. There are inequities in information with some countries such as Brazil having disproportionate representation in the literature. Additionally, certain themes such as prevalence, health outcomes and HIV risk factors are well represented, while important focuses such as interventions are neglected, with only three studies describing strategies to reduce HIV risk among PWUS in LA. The body of information presented here frames SUD as a major public health concern as it relates to PLWH; however, there is scant information regarding what does and does not work to fix the problem. Data assessing interventions are available for other regions; however, given the unique geographic and cultural contexts of LA, it is essential to perform similar research within this region. Though much progress has been made in highlighting that SU is a major concern as it relates to HIV, much more is yet to be done to work towards reducing HIV incidence in LA.

## Figures and Tables

**Figure 1 ijerph-19-07198-f001:**
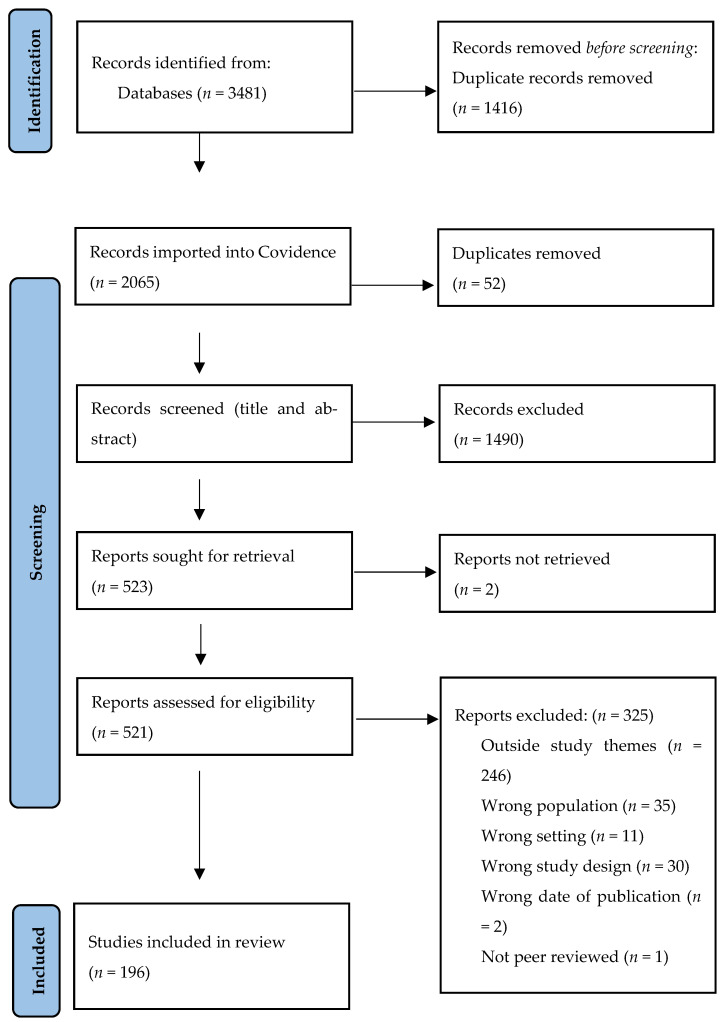
Prisma flowchart.

**Table 1 ijerph-19-07198-t001:** Number of papers per country.

Country	# of Papers
Brazil	127
Mexico	29
Colombia	17
Peru	16
Argentina	13
El Salvador	7
Chile	4
Honduras	3
Ecuador	3
Guatemala	3
Costa Rica	2
Uruguay	1
Nicaragua	1
Venezuela	1
Panama	1
Belize	1

Note: Sum of papers per country is not the same as the total number of papers included, because some papers were conducted in more than one country.

## Data Availability

Not applicable.

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
