# Peer review of "HIV and Substance Use in Latin America: A Scoping Review"

_ijerph, 2022, doi:10.3390/ijerph19127198_

Round 1

Reviewer 1 Report

The review summarizes the relationship between HIV and substance use (SU) in Latin America (LA). However, with unequal representation of countries in LA, most papers were from Brazil, with some papers from Peru, Mexico, Colombia, and Argentina, and the rest of the LA countries with one or zero papers. The authors screened 3,481 references and included 196 references in the review; of which, 83 studies form10 countries reported the prevalence of SU in people living with HIV/AIDS (PLWH) in Brazil (59), Argentina (5), Peru (5), Mexico (5), Colombia (2), Ecuador (1), Venezuela (1), Chile (1), Guatemala (1), and Uruguay (1). The paper summarized the prevalence of SU in PLWH and HIV in people who use substances (PWUS) and the factors associated with HIV positivity among PWUS; identified risky sexual behaviors among PLWH and PWUS and analyzed the strategies for reducing HIV risk in PWUS and SU risk in PLWH in LA. The review pictured the complex and bidirectional relationship between SU and HIV in LA; identified that SU is a major concern in PLWH; suggested that effective interventions are needed for reducing HIV incidence in LA.

Minor changes:

Line 28: Please provide the full name for PLWH 

Figure 1: Please adjust the text box size and make the font readable. Check the figure legend font.

Please use “Table” instead of “table” in the context to keep it consistent.

Line 166 states there are 2 papers about Colombia, however, there is only one reference under the Colombia section in Table 2.

Typos in Table 2:

Guatemala:  ”PLWH attending al Infectious Diseases clinic (Guatemala City)“  Is that “all”?

Multicountry:  (1.9-11-1%), is that "11.1%"

line 221: VL full name

Line 358: AUD full name

Please check the reference format keep it consistent: i.e. ref 165, 182, 186, 205, 221, 227, 229.

Author Response

  1. Line 28: Please provide the full name for PLWH:
    We have edited the text to provide the full name for PLWH.

  2. Figure 1: Please adjust the text box size and make the font readable. Check the figure legend font:
    We have edited the text box size to make the font readable and edited the figure legend font to fit the rest of the manuscript.

  3. Please use “Table” instead of “table” in the context to keep it consistent:
    We have edited the manuscript accordingly.

  4. Line 166 states there are 2 papers about Colombia, however, there is only one reference under the Colombia section in Table 2:
    Thank you for this observation. We have added the second reference for Colombia in Table 2.

  5. Typos in Table 2:
    1. Guatemala: “PLWH attending al Infectious Diseases clinic (Guatemala City)” Is that “all”?
      It should say “PLWH attending an Infectious Diseases clinic (Guatemala City)”. We have edited the text to reflect this.

    2. Multicountry: (1.9-11-1%), is that "11.1%”:
      It should say “11.1%”. We have edited the text to reflect this.

  6. Line 221: VL full name:
    We have edited the text to include “viral load”.

  7. Line 358: AUD full name:
    We have edited the text to include “alcohol use disorder”.

  8. Please check the reference format keep it consistent: i.e. ref 165, 182, 186, 205, 221, 227, 229:
    We have edited these references to keep them consistent with the formatting of the journal.

Reviewer 2 Report

Dear Authors,

This is a very important piece of work and you have addressed some issues in relation to substance use and HIV infections. I am happy with the PRISMA method you have used. 

I would like to see in the results and methods on study designs and their impact on results. Did all the studies use the same instruments to measure tobacco and drug use? 

Thanks

Author Response

  1. I would like to see in the results and methods on study designs and their impact on results. Did all the studies use the same instruments to measure tobacco and drug use?
    This is a very important research question, albeit one that goes beyond the scope of our study. We mentioned in the results section that not all studies used the same instruments to measure tobacco and drug use, and, as mentioned in the discussion, this complicates the comparison of ranges reported in the literature both within Latin America and with other regions. We make the suggestion that further research utilize the Pan American Health Organization and World Health Organization recommendations to improve comparability of research about substance use and abuse, to ensure that in the future we have more standardized data.

Reviewer 3 Report

The manuscript addresses an important health issue for Latin Americans - the relationship between substance abuse and HIV.   An appropriate framework, The Preferred Reporting Items for Systematic Reviews and Meta-Analyses (PRISMA-ScR), was selected.   The scoping review is comprehensive and soundly conducted overall.  Below are some recommendations for refinement of the manuscript:

(1) Be sure to define all terms when they are used for the first time.  For example, the term PLWH is used in the Abstract in line 28 

(2) Inclusive Definition of Substance Use - The implications of combining substances of various types in measuring disease prevalence addressed adequately in terms of their impact on the results.  However, since a statement is made that most reviewed studies did not follow Pan American Health Organization and World Health Organization recommendations for improving comparability of research (lines 385-396), a more thorough discussion is needed in that paragraph. Because the article is written as a scoping review, the authors should clarify or elaborate on their statement in lines 394-396 "Without quality data, it becomes difficult to understand the scope of the problem, develop interventions and measure their success." 

(3) Balance - In Table 1 it appears as though a high proportion of the sum of papers include Brazil. Additional discussion of vertical transmission, as well as the higher level of risk of HIV conferred by IDU compared to non-IDU, section 3.3 lines 218-230 of the manuscript would be helpful.  Also, since sex work was a mediating variable on the association between gender and HIV incidence, additional commentary could be provided in lines 210-212.

(4) Strategies for reducing SU risk in PLWH in LA.   Given that only one cohort study was conducted among TGW initiating ART in a trans-sensitive clinic in Argentina without a control group - there may be an overemphasis in the manuscript on trans-sensitive healthcare as a major strategy (353-355).  Consider moving this down on the list of strategies, or identifying the limitations more strongly in the discussion section.   

(5) Discussion - In lines 423-425 it is stated that only three papers included in the review described strategies to reduce HIV risk among PWUS in LA.   The limitations should be elaborated upon in the first paragraph of the Conclusions sections 

Author Response

  1. Be sure to define all terms when they are used for the first time. For example, the term PLWH is used in the Abstract in line 28:
    Thank you for this observation. We have defined PLWH in the abstract, as well as viral load (VL) and alcohol use disorder (AUD) in the main text.

  2. Inclusive Definition of Substance Use - The implications of combining substances of various types in measuring disease prevalence addressed adequately in terms of their impact on the results. However, since a statement is made that most reviewed studies did not follow Pan American Health Organization and World Health Organization recommendations for improving comparability of research (lines 385-396), a more thorough discussion is needed in that paragraph:
    Thank you for this suggestion. We have expanded the discussion on this topic in the manuscript.

  3. Because the article is written as a scoping review, the authors should clarify or elaborate on their statement in lines 394-396 "Without quality data, it becomes difficult to understand the scope of the problem, develop interventions and measure their success.":
    Thank you for this comment. We have decided to edit this statement and instead of using the phrase “the scope of the problem”, use the phrase “the burden of the problem”. This scoping review highlights the need for standardizing definitions and methodologies to ensure comparable data.

  4. Balance - In Table 1 it appears as though a high proportion of the sum of papers include Brazil. Additional discussion of vertical transmission, as well as the higher level of risk of HIV conferred by IDU compared to non-IDU, section 3.3 lines 218-230 of the manuscript would be helpful. Also, since sex work was a mediating variable on the association between gender and HIV incidence, additional commentary could be provided in lines 210-212:
    We have included additional discussion about vertical transmission of HIV, higher levels of HIV conferred by IDU as compared to non-IDU, and the association between gender and HIV incidence.

  5. Strategies for reducing SU risk in PLWH in LA. Given that only one cohort study was conducted among TGW initiating ART in a trans-sensitive clinic in Argentina without a control group - there may be an overemphasis in the manuscript on trans-sensitive healthcare as a major strategy (353-355). Consider moving this down on the list of strategies or identifying the limitations more strongly in the discussion section:
    We have moved this section down on the list of strategies, as suggested, and added a sentence in the discussion section to clarify that the mentioned study has a cohort design.

  6. Discussion - In lines 423-425 it is stated that only three papers included in the review described strategies to reduce HIV risk among PWUS in LA. The limitations should be elaborated upon in the first paragraph of the Conclusions sections
    Thank you for this comment. We have elaborated upon this limitation in the Conclusions section.